# Inducing and tuning Kondo screening in a narrow-electronic-band system

Shiwei Shen [1], Chenhaoping Wen [1], Pengfei Kong[1], Jingjing Gao[2,3], Jianguo Si[2], Xuan Luo [2], Wenjian Lu [2], Yuping Sun [2,4,5], Gang Chen[6] & Shichao Yan [1,7 ✉]

Although the single-impurity Kondo physics has already been well understood, the understanding of the Kondo lattice where a dense array of local moments couples to the conduction electrons is still far from complete. The ability of creating and tuning the Kondo lattice in non-*f*-electron systems will be great helpful for further understanding the Kondo lattice behavior. Here we show that the Pb intercalation in the charge-density-wave-driven narrow-electronic-band system 1*T*-TaS$_2$ induces a transition from the insulating gap to a sharp Kondo resonance in the scanning tunneling microscopy measurements. It results from the Kondo screening of the localized moments in the 13-site Star-of-David clusters of 1*T*-TaS$_2$. As increasing the Pb concentration, the narrow electronic band derived from the localized electrons shifts away from the Fermi level and the Kondo resonance peak is gradually suppressed. Our results pave the way for creating and tuning many-body electronic states in layered narrow-electronic-band materials.

[1] School of Physical Science and Technology, ShanghaiTech University, Shanghai 201210, China. [2] Key Laboratory of Materials Physics, Institute of Solid State Physics, HFIPS, Chinese Academy of Sciences, Hefei 230031, China. [3] University of Science and Technology of China, Hefei 230026, China. [4] High Magnetic Field Laboratory, HFIPS, Chinese Academy of Sciences, Hefei 230031, China. [5] Collaborative Innovation Centre of Advanced Microstructures, Nanjing University, Nanjing 210093, China. [6] Department of Physics and HKU-UCAS Joint Institute for Theoretical and Computational Physics at Hong Kong, The University of Hong Kong, Hong Kong, China. [7] ShanghaiTech Laboratory for Topological Physics, ShanghaiTech University, Shanghai 201210, China. ✉email: yanshch@shanghaitech.edu.cn

In the narrow-electronic-band or even the flat-band-like systems, the electron kinetic energy is strongly suppressed, and the electron correlation is then enhanced. When the electron correlation energy becomes comparable to the kinetic energy, the system is strongly correlated. This makes the narrow-electronic-band system a versatile platform for exploring the exotic correlated electronic phases. Well-known examples include the narrow band of $f$-like electrons with Kondo lattice and heavy fermions[1], and recently, the magic-angle twisted bilayer graphene superlattices with Mott-insulating gap and topological electronic phases[2–4].

In layered narrow-electronic-band materials, the Coulomb interaction and interlayer coupling often complex their electronic states. One of the notable examples is the charge density wave (CDW) driven narrow-electronic-band material, $1T$-TaS$_2$. $1T$-TaS$_2$ is a layered transition metal dichalcogenide (TMD) and undergoes a commensurate CDW transition around 180–190 K[5–7]. In this commensurate CDW state of $1T$-TaS$_2$, the in-plane lattice distortion leads to the formation of Star-of-David (SD) cluster which consists of 13 Ta atoms, and there is one unpaired electron in each SD cluster. Further band structure calculations indicate the unpaired electrons in the SD clusters form a narrow electronic band that may be susceptible to the Mott-Hubbard transition by the in-plane Coulomb interaction[8]. The resulting Mott insulating state is quoted as a cluster Mott insulator where the unpaired electron is localized on the SD cluster rather than on the original Ta lattice site. In the cluster Mott insulating state, this unpaired electron forms an effective spin-1/2 local magnetic moment. No conventional magnetic ordering has been detected in $1T$-TaS$_2$ down to millikelvin temperature, and $1T$-TaS$_2$ has been proposed to be a spin-1/2 triangular lattice quantum spin liquid candidate[9]. This local moment formation and spin liquid scenario have been supported by the recent muon spin relaxation measurement in $1T$-TaS$_2$[10] and the scanning tunneling microscopy (STM) measurement in the epitaxially-grown isostructural material (monolayer $1T$-TaSe$_2$)[11].

Apart from the novel Mott localization of the SD cluster in $1T$-TaS$_2$, the previous theoretical and experimental studies indicate that the interlayer stacking effect plays a crucial role in the electronic properties of $1T$-TaS$_2$[12–16]. Due to the interlayer coupling in $1T$-TaS$_2$, the interlayer stacking pattern of the SD clusters yields two distinct surface terminations[13,17]. The Type-I surface terminates at the end of the dimerization, while the Type-II surface terminates in the middle of the dimerization and thus cuts the dimerization (see Fig. 1a). In the differential conductance (d$I$/d$V$) spectra measured with STM, there are insulating gaps for both surface terminations, and the Type-I surface has a slightly larger insulating gap than that on the Type-II surface[13]. It has recently been demonstrated that the insulating gap on the Type-I surface is an interlayer-dimerization-induced insulating gap and the smaller insulating gap on the Type-II surface is likely a Mott-insulating gap[18,19].

More surprisingly, the very recent study demonstrates that when the monolayer $1T$-TaS$_2$ is grown on the monolayer metallic $1H$-TaS$_2$ by molecular beam epitaxy, the Kondo resonance peak appears in the $1T$-TaS$_2$ layer[20], which confirms the local moment formation and possible cluster Mott physics in the monolayer $1T$-TaS$_2$. If the insulating gap of the Type-II surface in the bulk $1T$-TaS$_2$ is truly a Mott-insulating gap, there should be local magnetic moment formation that is similar as the monolayer $1T$-TaS$_2$. The nature of a band-insulating gap instead of a Mott-insulating gap would indicate the absence of such a magnetic moment formation. To address this issue, the Kondo resonance could be an ideal experimental diagnosis of the local moment formation on the Type-II $1T$-TaS$_2$ surface. Here we report a low-temperature and high-magnetic-field STM study about $1T$-TaS$_2$.

We observe the insulating gap to Kondo resonance transition on the Type-II $1T$-TaS$_2$ surface by the Pb intercalation. We further establish the relationship between the narrow electronic band and the Kondo resonance peak. These observations are helpful for understanding the origin of the insulating gap on the Type-II $1T$-TaS$_2$ surface and the Kondo resonance behavior in the narrow-electronic-band systems.

## Results and discussion

After cleaving the $1T$-TaS$_2$ sample in ultrahigh vacuum, we evaporate the Pb atoms onto the cleaved $1T$-TaS$_2$ sample. We find both the Type-I and Type-II surface terminations exist. In Fig. 1d, g, we depict the STM topographies taken on the Type-I and Type-II surface terminations of $1T$-TaS$_2$. The Pb atoms form regular islands on these two surfaces (the detailed structure of the Pb islands are shown in Supplementary Figs. S1, S2). However, the bare $1T$-TaS$_2$ regions of the two surfaces appear quite different. The bare $1T$-TaS$_2$ of the Type-I surface is the same as the pristine $1T$-TaS$_2$ surface with uniform SD clusters and a few intrinsic atomic defects (see Fig. 1d). However, in the bare $1T$-TaS$_2$ region of the Type-II surface, except the intrinsic atomic defects, some of the SD clusters appear as dark spots (see Fig. 1g). These dark SD clusters are almost indistinguishable in STM topographies taken with positive bias voltages (Supplementary Fig. S3), and the position of the dark SD cluster can be changed but rarely during the STM measurements (Supplementary Fig. S4). The boundary between these two types of the surfaces can also be seen on the same $1T$-TaS$_2$ sample (Supplementary Fig. S5). In comparison with the SD clusters on the bare Type-I surface, we conclude that for the Type-II surface there are intercalated Pb atoms beneath the dark SD clusters and no intercalated Pb atoms under the bright SD clusters (Supplementary Fig. S5). Our STM data indicate that Pb atoms are prone to intercalate into the van der Waals gap below the Type-II surface. This is likely because the van der Waals gap below the Type-II surface is slightly larger than that below the Type-I surface[13].

In Fig. 1e, h, we show the zoom-in STM topographies taken on the two different surface terminations. Figure 1f, i are the differential conductance (d$I$/d$V$) spectra taken on the Pb island and the bare $1T$-TaS$_2$ region for the two surface terminations, respectively. As shown in Fig. 1f, i, the Pb islands on the two surfaces are both metallic (see the green spectra). The d$I$/d$V$ spectrum taken on the bare $1T$-TaS$_2$ region of the Type-I surface is the same as that taken on the pristine Type-I surface, where the ~150 mV insulating gap can be seen (the purple spectrum shown in Fig. 1f). In contrast, instead of showing a Mott-insulating gap as on the pristine Type-II surface, a pronounced zero bias peak (ZBP) appears in the d$I$/d$V$ spectrum taken on the bare Type-II surface (the purple spectrum shown in Fig. 1i). Although the dark SD clusters appear in most of the bare Type-II regions, we can still find a small region on the bare $1T$-TaS$_2$ surface where there are no dark SD clusters (as shown by the dashed black lines in Fig. 2a). In this small region, the ~50 mV insulating gap as shown on the pristine Type-II surface can be seen (Fig. 2b and Supplementary Fig. S6)[13]. This demonstrates that the strong ZBP shown in Fig. 1i is indeed induced by the intercalation of Pb atoms below the Type-II surface. As shown by the yellow circles in Fig. 2a, there is no CDW domain wall in the Pb intercalated region, and the SD clusters in the Pb intercalated region and the ~50 mV gap region are also well aligned. This indicates that the intercalated Pb atoms do not create CDW domain walls (see Supplementary Fig. S7 for more details), which is different from the Se substituted $1T$-TaS$_2$ and copper intercalated $1T$-TiSe$_2$ where the substituted or intercalated atoms create CDW domain walls[21,22]. We note that on the Type-II surface, we also observe a few

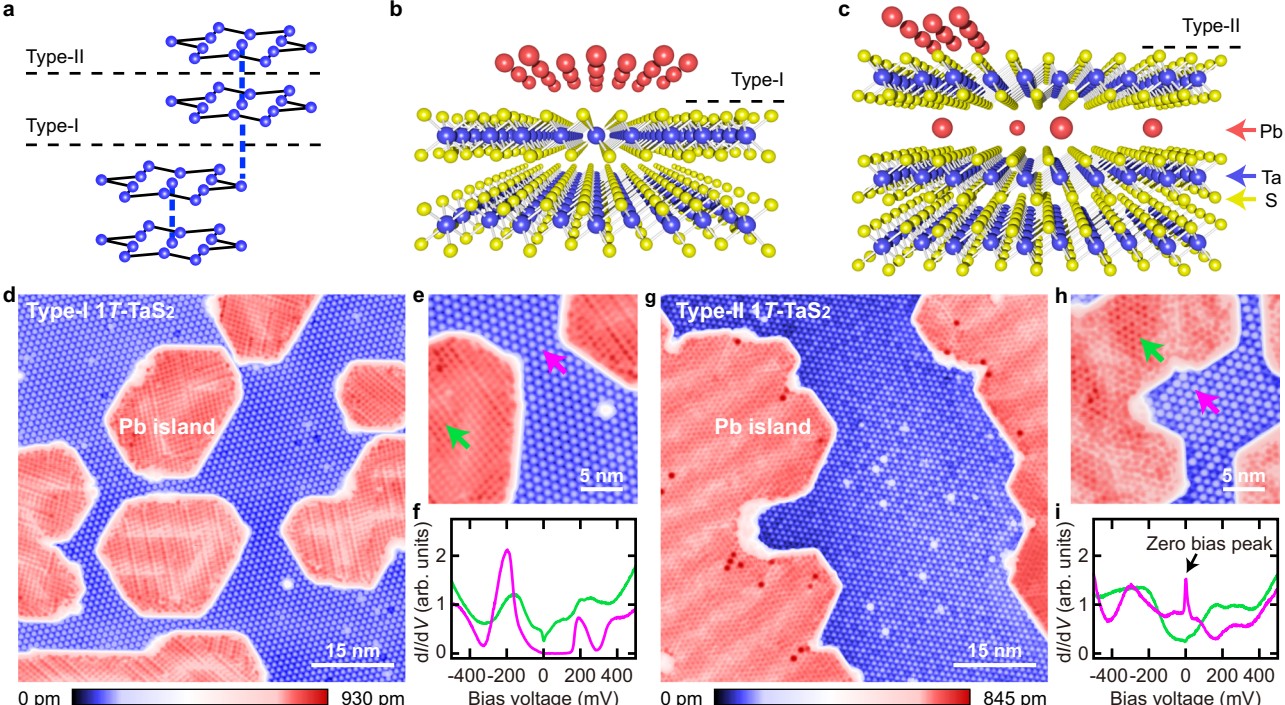

**Fig. 1 Emergence of the zero bias peak in the Pb intercalated 1T-TaS₂.** **a** Schematic showing the alternating interlayer stacking of the SD clusters in 1T-TaS₂ which results in two inequivalent cleavage planes. **b** Schematic for the structure of the Type-I surface after evaporating Pb atoms. **c** Schematic for the structure of the Type-II surface after evaporating Pb atoms. The red, blue and yellow balls in **b**, **c** represent the Pb, Ta and S atoms, respectively. **d** Constant-current STM topography taken on the Type-I surface ($Vs = -500$ mV, $I = 20$ pA). **e** Zoom-in view of the bare Type-I surface and Pb islands ($Vs = -500$ mV, $I = 20$ pA). **f** Differential conductance (d$I$/d$V$) spectra taken on the spots indicated by the purple and green arrows shown in **e**. **g** Constant-current STM topography taken on the Type-II surface ($Vs = -500$ mV, $I = 30$ pA). **h** Zoom-in view of the bare Type-II surface and Pb islands ($Vs = -500$ mV, $I = 20$ pA). **i** d$I$/d$V$ spectra taken on the spots indicated by the purple and green arrows in **h**. The data shown in this figure are taken at $T = 4.3$ K. Source data are provided as a Source Data file.

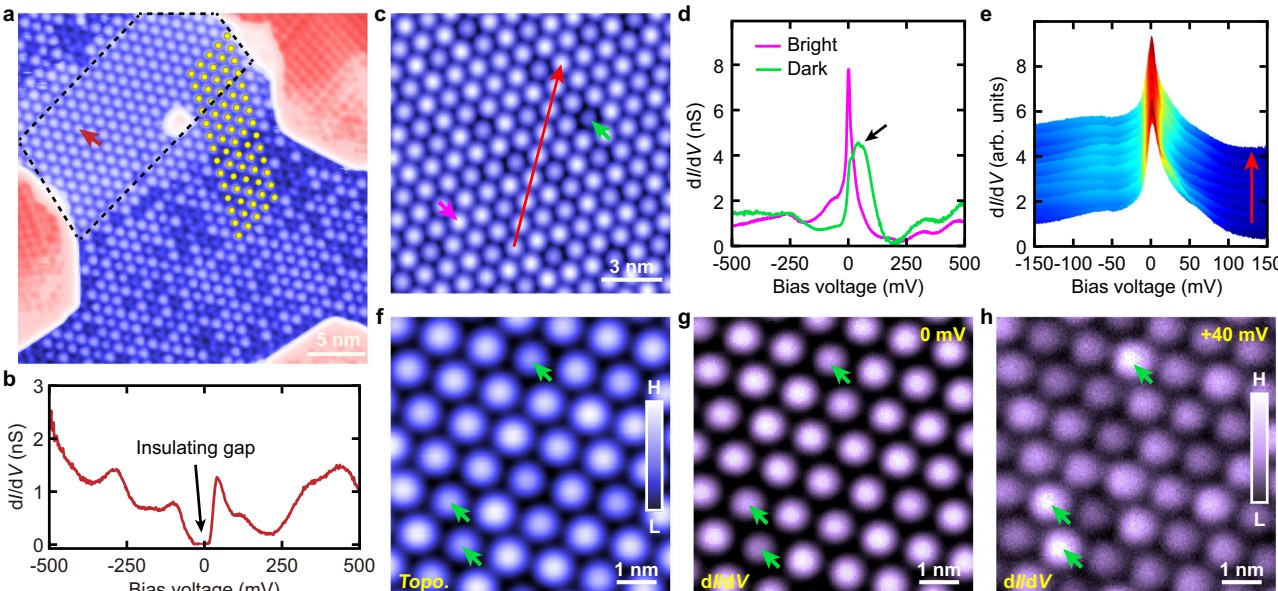

**Fig. 2 Spatial distribution of the ZBP.** **a** Constant-current STM topography taken on the Type-II 1T-TaS₂ surface ($Vs = -500$ mV, $I = 20$ pA). The yellow circles indicate the centers of the SD clusters. The dashed black lines show the region without the dark SD clusters. **b** d$I$/d$V$ spectrum taken on the spot shown by the red arrow in **a**. **c** Constant-current STM topography taken on the Type-II surface with dark SD clusters ($Vs = -150$ mV, $I = 30$ pA). **d** d$I$/d$V$ spectra taken on the bright and dark SD clusters shown by the purple and green arrows in **c**. The black arrow indicates the electronic state located slightly above the Fermi level in the dark SD cluster. **e** Spatially distributed d$I$/d$V$ spectra along the red arrow in **c**. **f** Constant-current STM topography taken on the Type-II surface with dark SD clusters ($Vs = -200$ mV, $I = 30$ pA). **g**, **h** d$I$/d$V$ maps taken on the same region as shown in **f** with 0 mV (**g**) and +40 mV (**h**) bias voltages, respectively. The green arrows in **f**–**h** indicate the positions of the dark SD clusters. The data shown in this figure are taken at $T = 4.3$ K. Source data are provided as a Source Data file.

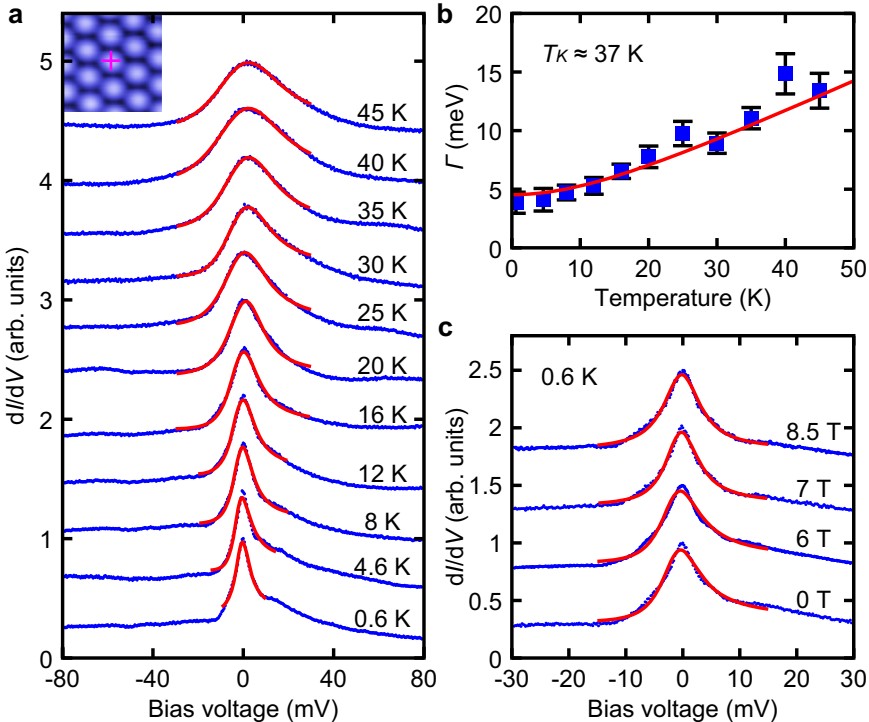

**Fig. 3 Temperature and magnetic field dependences of the ZBP. a** The blue lines are the d$I$/d$V$ spectra measured at different temperatures on the bright SD cluster of the Type-II 1$T$-TaS$_2$ surface. The red lines are the thermally convolved Fano fits to the ZBP at each temperature. The purple cross in the inset shows the position for taking the temperature dependent d$I$/d$V$ spectra. The spectra are vertically offset for clarity. **b** Evolution of the measured half width ($\Gamma$) at half maximum of the ZBP with temperature (blue squares with error bar). The error bars represent the uncertainty of the linewidths obtained from the thermally convolved Fano fits to the ZBP at each temperature. The solid red line is the best fit to the function $\sqrt{(\pi k_B T)^2 + 2(k_B T_K)^2}$, which yields a Kondo temperature of $T_K \approx 37 \pm 6$ K. **c** The blue lines are the d$I$/d$V$ spectra measured with different external magnetic fields at 0.6 K. The red lines are the thermally convolved Fano fits to the ZBP at the measured magnetic fields. The spectra are vertically offset for clarity. Source data are provided as a Source Data file.

individual Pb adatoms, and there is no ZBP in the d$I$/d$V$ spectra taken on the Pb adatoms (Supplementary Fig. S8).

In order to understand the nature of the ZBP, we perform the spatially-resolved d$I$/d$V$ measurements on the dark SD clusters and the bright SD clusters on the Type-II surface. As shown in Fig. 2d, the strong ZBP appears on the bright SD cluster (shown with the purple arrow in Fig. 2c). On the dark SD cluster (shown with the green arrow in Fig. 2c), the intensity of the ZBP is strongly suppressed and a new electronic peak located slightly above the Fermi level emerges (shown with the black arrow in Fig. 2d). The d$I$/d$V$ linecut along the red arrow in Fig. 2c indicates the ZBP persist in every bright SD cluster (Fig. 2e and Supplementary Fig. S9). Figure 2g, h are the d$I$/d$V$ maps taken on the same region as shown in Fig. 2f, which clearly show the spatial distributions of the electronic states on the Pb intercalated Type-II surface. As we can see, the ZBP peak has higher intensity at the center of each bright SD cluster (Fig. 2g). The +40 mV electronic state locates at the dark SD clusters (shown by the green arrows in Fig. 2h). Since the Type-II surface is an undimerized 1$T$-TaS$_2$ surface where there is an unpaired electron localized in each SD cluster, the ZBP is likely due to the Kondo screening of the unpaired electrons in the SD clusters.

To further confirm the Kondo nature of the ZBP, we perform the temperature and magnetic field dependent d$I$/d$V$ measurements. In order to reduce the broadening effect induced by the modulation voltage, we reduce the modulation voltage to be 0.2 mV. As shown in Fig. 3a, the ZBP at each temperature is fitted reasonably well with the thermally convolved Fano line shape with intrinsic width[11,23]. Fitting the half width at half maximum

of the temperature-dependent ZBP with the well-known Kondo expression yields a Kondo temperature $T_K \sim 37$ K (Fig. 3b)[23,24]. In addition, we perform the magnetic field dependent d$I$/d$V$ measurements at 0.6 K. As shown in Fig. 3c, no clear splitting of the Kondo resonance peak is observed with external magnetic field up to 8.5 T. However, the Kondo resonance peak in the single layer 1$T$-TaS$_2$ is clearly split with 10 T magnetic field[20]. The difference is that the width of Kondo resonance peak in Fig. 3c is about four times the width of the Kondo resonance peak in the single layer 1$T$-TaS$_2$ (Supplementary Fig. S10). This makes the Kondo resonance peak in Fig. 3c less sensitive to the splitting induced by the magnetic field, and larger magnetic field is needed to induce the splitting of the Kondo resonance on the Pb intercalated Type-II 1$T$-TaS$_2$ surface[25,26]. Similar effect has also been observed in the heavy fermion metal where the Kondo lattice peak has no clear splitting with magnetic field up to 11 T[27].

Having identified the Kondo nature of the ZBP, we next discuss the origin of the electronic state located slightly above the Fermi level in the dark SD cluster marked by the green arrow in Fig. 2c. In the CDW state of 1$T$-TaS$_2$, the unpaired electrons in each SD cluster form a narrow electronic band near the Fermi level[8]. The new electronic band at the dark SD cluster is the narrow electronic states derived from the unpaired electrons in the SD clusters. This also indicates that the intercalated Pb atom depletes the localized electron in the dark SD cluster, which makes the narrow electronic band almost unoccupied and locate slightly above the Fermi level.

The next question is why the intercalated Pb atoms can induce the transition from the ~50 mV insulating gap to the sharp

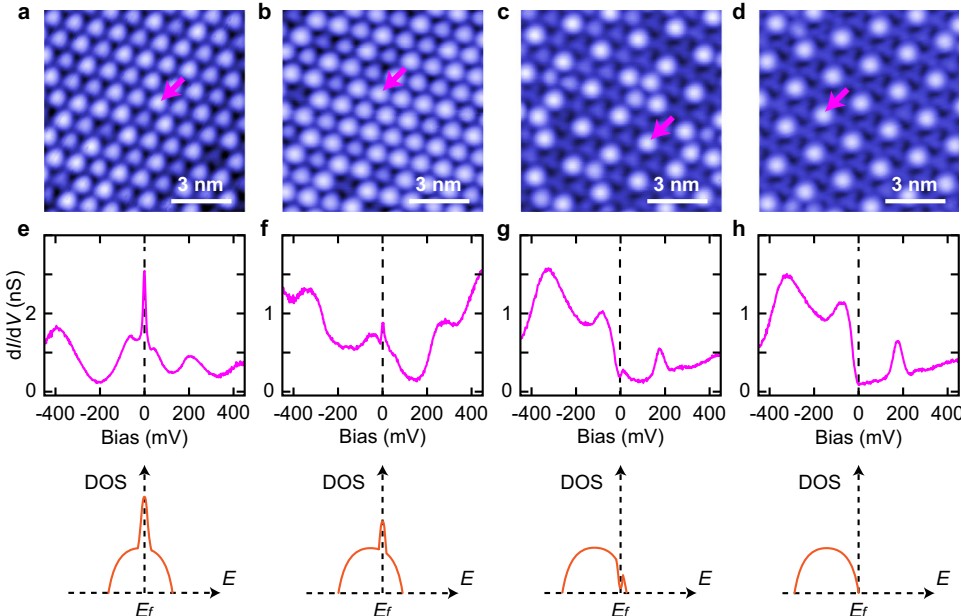

**Fig. 4 Evolution of the Kondo resonance peak. a–d** Constant-current STM topographies taken on the Type-II 1$T$-TaS$_2$ surface as increasing the intercalant concentration of Pb ($Vs = -500$ mV, $I = 20$ pA). **e–h** Top panels: d$I$/d$V$ spectra taken on the purple arrows shown in **a–d**, respectively. Bottom panels: schematics showing the evolution of the narrow electronic band and Kondo resonance peak shown in the top panels. The data shown in this figure are taken at $T = 4.3$ K. Source data are provided as a Source Data file.

**Kondo resonance peak on the Type-II 1$T$-TaS$_2$ surface.** Recently, motivated by the Kondo resonance in single layer 1$T$-TaSe$_2$ grown on single layer 1$H$-TaSe$_2$[11], Chen *et al.* have developed a model for the correlated electrons coupled by tunneling to a layer of itinerant metallic electrons to explain the transition to the Kondo resonance[28]. We think the insulating gap to Kondo resonance transition on Type-II surface can also be explained by the model developed by Chen et al.[28] The ~50 mV insulating gap on the pristine Type-II surface is previously interpreted as a Mott-insulating gap that is induced by the in-plane Coulomb interaction[19]. Without the Pb intercalation, the unpaired localized electrons in the SD clusters of the Type-II surface weakly couples with the underneath insulating dimerized 1$T$-TaS$_2$ layer that has few itinerant electrons. Upon the intercalation of the Pb atoms, there is charge transfer between the intercalated Pb atoms and the host 1$T$-TaS$_2$ layer (Supplementary Fig. S11)[29,30]. Giving the picture of local magnetic moment formation in the topmost 1$T$-TaS$_2$ layer, the more itinerant electrons from the Pb intercalation underneath the top layer couple to the Mott-localized electron in the topmost 1$T$-TaS$_2$ layer. Since this topmost 1$T$-TaS$_2$ layer is in the Mott regime, the effect would be of the second order, i.e., the Kondo exchange coupling between the itinerant electrons and the local moments in the topmost 1$T$-TaS$_2$ layer[28]. This Kondo exchange is the key to generate the sharp Kondo resonance peak in the d$I$/d$V$ spectrum. According to the temperature-dependent Kondo resonance width shown in Fig. 3b and the model developed by Chen et al. in ref. [28], the situation after Pb intercalation should be either near the localized Kondo limit or in the regime where the Kondo exchange coupling is comparable with the Heisenberg exchange coupling between the local moments in the 1$T$-TaS$_2$ layer[28].

The situation here can be considered analogous to the heavy fermion systems. The narrow electronic band formed by the unpaired electrons in the SD clusters localizes the electronic wave functions in place of the heavy $f$ electrons, and the itinerant electrons are from the intercalated Pb atoms and the underneath 1$T$-TaS$_2$ layers. More interestingly, we find that the intensity of

the Kondo resonance peak in the bright SD clusters is strongly related with the energy position of the narrow electronic band formed by the unpaired electrons in the SD clusters. As shown in Fig. 4a–d (see Supplementary Fig. S12 for more details), when the intercalant concentration of the Pb atoms increases, the narrow electronic band is gradually shifted to be below the Fermi level (Fig. 4e–h). At the same time, the Kondo peak is gradually reduced (Fig. 4e, f), and it can be fully suppressed (Fig. 4g, h). The evolution of d$I$/d$V$ spectra in the dark SD clusters are shown in Supplementary Fig. S13. This means that as increasing the Pb concentration, the narrow electronic band is gradually electron doped at the positions of the bright SD clusters, and the Kondo resonance peak is related with its filling factor.

Furthermore, we find that as increasing the intercalant concentration of Pb, the dark and bright SD clusters on the Type-II surface form a new $\sqrt{3} \times \sqrt{3}$ superstructure with respect to the SD lattice (Fig. 5a). The green and purple dots in Fig. 5a indicate the dark and bright SD clusters. Figure 5b is the STM topography with positive bias voltage taken on the same region as Fig. 5a, where the contrast between the dark and bright SD clusters is weaker and the individual SD clusters can be clearly seen. As shown in the d$I$/d$V$ spectra taken on the bright and dark SD clusters (Fig. 5c), the narrow electronic band in the dark SD cluster is fully unoccupied (the green spectrum), and it is fully occupied in the bright SD cluster (the purple spectrum). Figure 5d–f show the spatial distributions of the electronic states at the energies indicated by the vertical dashed lines in Fig. 5c (More d$I$/d$V$ maps are shown in the Supplementary Fig. S14). As we can see, the $-80$ mV and the $+40$ mV states show contrast inversion, and they have higher intensity in the bright and dark SD clusters, respectively. Our data indicate the intercalated Pb atoms make the localized electrons in the SD clusters of the top 1$T$-TaS$_2$ layer to be spatially redistributed and the new $\sqrt{3} \times \sqrt{3}$ superstructure results from the spatial charge modulation of the localized electrons. In this case, the narrow electronic band in the bright SD clusters is locally electron doped and it corresponds to local hole doping in the dark SD clusters (insets in Fig. 5c).

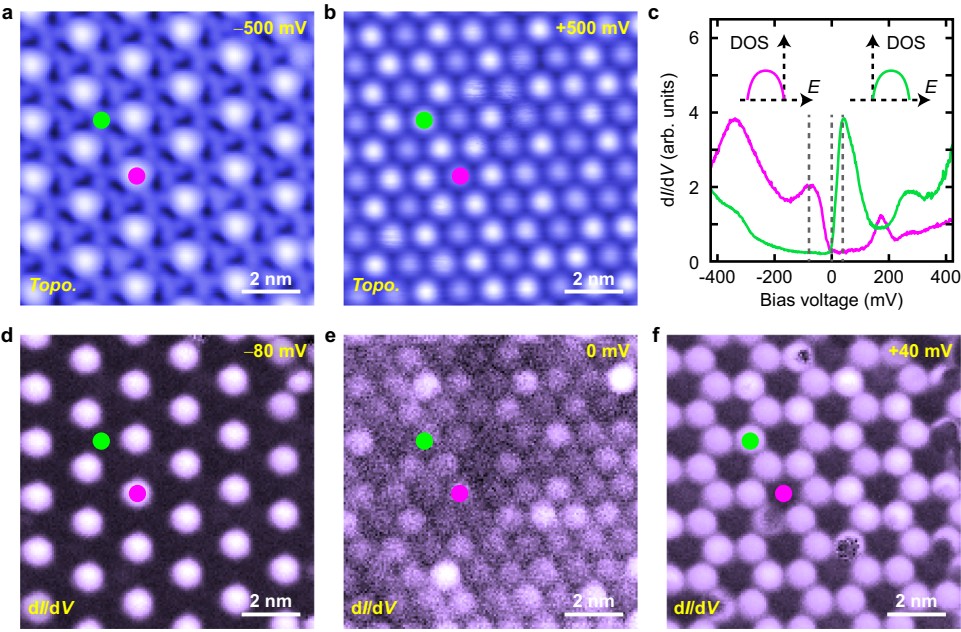

**Fig. 5 Electronic states in the $\sqrt{3} \times \sqrt{3}$ superstructure. a** Constant-current STM topography taken on the Type-II 1$T$-TaS$_2$ surface with negative bias voltage ($Vs = -500$ mV, $I = 20$ pA). **b** Constant-current STM topography taken with positive bias voltage on the same region as **a** ($Vs = +500$ mV, $I = 20$ pA). **c** d$I$/d$V$ spectra taken on the purple and green dots shown in **a**, **b**. The insets show the energy positions of the narrow electronic band on the purple and green dots shown in **a**, **b**. **d–f** d$I$/d$V$ maps taken on the same region as shown in **a**, **b** with −80 mV (**d**), 0 mV (**e**) and +40 mV (**f**) bias voltages, respectively. The data shown in this figure are taken at $T = 4.3$ K. Source data are provided as a Source Data file.

Our work demonstrates that the intercalated Pb atoms below the undimerized 1$T$-TaS$_2$ layer induce the insulating gap to Kondo resonance transition. The strength of the Kondo resonance peak can be deliberately tuned by the intercalant concentration of Pb. Our data not only demonstrate the interplay between the narrow electronic band and the Kondo resonance peak, but also shed new light on the origin of the insulating gap in the undimerized 1$T$-TaS$_2$ layer. Our results should be helpful for understanding the Kondo lattice behavior in heavy fermion materials[1] and the many-body resonance in kagome metals[31]. We believe this method should be able to be extended to other foreign atoms or molecules to induce many-body electronic states in layered narrow-electronic-band materials.

## Methods

**Sample preparation**. 1$T$-TaS$_2$ samples were cleaved at room temperature and in an ultrahigh-vacuum chamber. After cleaving, Pb atoms were deposited onto the 1$T$-TaS$_2$ sample from a Knudsen cell. During dosing Pb, the 1$T$-TaS$_2$ sample was kept at room temperature, and the temperature of the Pb source was 470 °C. After dosing Pb, the 1$T$-TaS$_2$ sample was kept at room temperature for 5 to 10 min and then inserted into the STM head for measurements. The intercalant concentration of the Pb atoms below the Type-II surface was related with the room-temperature annealing time after dosing Pb (see Supplementary Fig. S12).

**STM measurements**. The STM experiments at 4.3 K were conducted with a Unisoku ultrahigh-vacuum and low-temperature STM. The variable-temperature and the magnetic-field-dependent STM experiments were carried out in a home-built He-3 STM. The tungsten tips were flashed by electron-beam bombardment for several minutes before use. The differential conductance was measured using a standard lock-in detection technique.

## Data availability

The data that support the findings presented here are available from the corresponding author upon reasonable request. Source data are provided with this paper.

## Code availability

The codes for fitting the data are available from the corresponding author upon reasonable request.

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

## Acknowledgements

The authors thank Profs. Fuchun Zhang, Patrick Lee, Xi Chen and Jianpeng Liu for fruitful discussions. S.Y. acknowledges the financial support from National Science Foundation of China (Grant No. 11874042), the National Key R&D program of China (Grant No. 2020YFA0309602) and the start-up funding from ShanghaiTech University. C.W. acknowledges the support from National Natural Science Foundation of China (Grant No. 12004250) and the Shanghai Sailing Program (Grant No. 20YF1430700). G.C. thanks the support from National Science Foundation of China with Grant No. 92065203, the Ministry of Science and Technology of China with Grants No. 2018YFE0103200, by the Shanghai Municipal Science and Technology Major Project with Grant No.2019SHZDZX04, and by the Research Grants Council of Hong Kong with General Research Fund Grant No. 17306520. J.G., X.L., J.S., W.L. and Y.S. thank the support of National Key Research and Development Program under Contract No. 2016YFA0300404, the National Nature Science Foundation of China under Contracts No. 11674326 and No. 11774351, and the Joint Funds of the National Natural Science Foundation of China and the Chinese Academy of Sciences' Large-Scale Scientific Facility under Contracts No. U1832141, No. U1932217 and U2032215.

## Author contributions

S.Y. conceived the experiment. S.S., C.W. and P.K. carried out the experiments and performed the analysis. J.G., L.X. and Y.S. were responsible for the sample growth. J.S. and W.L. performed the first-principles calculations. S.S., G.C. and S.Y. wrote the manuscript with the input from all authors.

## Competing interests

The authors declare no competing interests.
