## [Peer Review File · Nature Communications]

REVIEWER COMMENTS

Reviewer #1 (Remarks to the Author):

In the present manuscript, Shen et al. investigate the lattice Kondo problem by studying the coupling between localized and itinerant electrons in Pb-intercalated correlated insulator 1T-TaS₂. The authors exploit novel Pb intercalation technique to tune the surface layer gradually between insulating and metallic states. In the metallic state, they observe the zero-bias peak (ZBP) in tunneling spectra, which they associate with the Kondo resonance by analyzing its behavior with temperature and magnetic field. The authors use these signatures to prove the local magnetic moment formation on the 1T-TaS₂ surface, and thus support the Mott gap and quantum spin liquid scenarios widely debated recently. These finding also shed light onto the novel many-body regimes associated with the spin degree of freedom. The experimental results presented in the manuscript are convincing, of very high quality and obtained at high technical level. They will certainly be of great interest to the broad community working in the areas of correlated systems and material science.

While I find the results novel and interesting, the manuscript in its present version lacks important pieces of analysis and has somewhat convoluted message. Major revision is necessary to consider it for publication in Nature Communications. The detailed comments are provided below.

Major concerns:

1) The manuscript can benefit from a clear picture of the transition from a Mott state to a narrow-band state and Kondo-resonance therein that is suggested to occur in Type-II-terminated surfaces.

1a. Does the Mott state collapse under intercalation? If yes, which process is the driving force – bandwidth, doping, charge transfer etc.?

1b. Are the itinerant CDW states affected by intercalation? Here I would like to point the authors' attention to the paper by Qiao et al. Phys. Rev. X 7, 041054 (2017) and especially the green curve in Fig. 4e that seems comparable to that seen in dark clusters in the present study.

Could the orbital interplay suggested there be responsible for the Mottness collapse in the present case?

I would suggest measuring spectroscopic maps at energies corresponding to the expected itinerant band energies to address these questions.

2) The authors do not provide enough evidence that the dark clusters in Type-II termination are the ones with Pb atom beneath. The arguments provided in the Supplementary Section 3 are logically incomplete. For example, some clusters in Fig. 2 are have spectral intensity somewhere between the bright and dark ones. In my opinion, the smoking-gun evidence would be the tip-induced manipulation of the intercalant atom. Can gentle bias pulsing change the position or appearance of the dark clusters?

3) The authors should identify what changes to the long-range charge order are introduced by the Pb intercalation/deposition to put this study into context. Fourier transforms of the topographic images would be very helpful here.

I could not but notice the robustness of the uniform $\sqrt{13} \times \sqrt{13}$ state to the small levels of intercalation, which is seen also in doped and chemically substituted samples. Do the authors see the formation of domain structures or individual domain walls as well (Fig. S3a maybe)? It would be interesting to match this behavior to other notable cases of the intercalated TMDCs (e.g. $1T\text{-Cu}_x\text{TiSe}_2$) and to simulations of the doped Coulomb lattice gas (cf. Karpov and Brazovskii, *Sci. Rep.* 8, 4043 (2018) and Vodeb et al., *New J. Phys.* 21, 083001 (2019)).

4) In lines 103-104 the authors claim that the 150-mV insulating gap in Type-I termination is understood as the band gap, since Pb intercalation does not affect it. However, no proof of intercalation in Type-I termination is provided, moreover the authors say the surface is indistinguishable from the pristine one. Therefore, the only safe conclusion so far is that intercalation does not occur on the Type-I-terminated surface, rather than the gap is of the band insulator.

5) The authors demonstrate that the magnetic field has no effect on the ZBP, consistent with the Kondo resonance picture. The message will be much stronger, if some effect is demonstrated for the narrow band in the dark cluster. Alternatively, an estimate of the expected magnetic field effect should be provided.

6) Could the authors comment on the spatial variation of the ZBP width (Fig. 2)? Furthermore, is the temperature dependence shown in Fig. 3a measured at the same spot?

Minor concerns:

1) The authors find Type-I and Type-II terminations in a room-temperature cleaved crystal, in contrast to the recipe and reasoning provided in Ref. 13. I believe this observation deserves a comment, at least in the supplement.

2) The authors nicely demonstrate the feasibility of the intercalation as the tuning knob in 1T-TaS₂. I believe a more detailed recipe should be provided such that others could exploit this technique as well.

3) Is there a way to quantify the local or mean intercalant concentration? Could the authors match the annealing time to intercalant concentration?

4) Intercalation conditions, at which the images shown in panels a-d in Figure 4, should be provided.

5) Could the authors comment on the very different appearance of the Pb islands on Type-I and Type-II terminations? Are there domain walls on Type-I termination? Could the periodic height modulation on Type-II termination be related to a strain pattern or reconstruction?

Technical comments:

a. Line 45: "...undergoes a commensurate CDW transition around 180K..." – I suggest using 180-190K, as this would be more consistent with the cited sources and literature data.

b. Line 53: "...emergent triangular lattice..." – what property is emergent here?

c. Line 58: reference 11 describes an epitaxially-grown monolayer of 1T-TaSe₂ with the physics quite different from the bulk 1T-TaSe₂.

d. Line 138: "...the dark SD cluster in Fig. 2c" – please, add 'green arrow' to enhance the readability of the paper.

Reviewer #2 (Remarks to the Author):

Shiwei et. al. studied the effect of Pb intercalation on bulk 1T-TaS₂ using STM and STS. They showed that for certain amount of Pb intercalation, the insulating gap in the pristine Type-II 1T-TaS₂ surface becomes a zero-bias peak in the dI/dV spectra, which is claimed to be a Kondo resonance peak.

My major concern is about the novelty of the work for the Nature Communications standards. Kondo resonance in Ta-based dichalcogenide such as 1T-TaSe₂ and 1T-TaS₂ have been reported recently as in Nat. Phys. 17, 1154 (2021) and arXiv:2103.11989 (2021). In addition, theoretical calculations to justify the main claim of the paper, such as the modeling of structural properties Pb intercalation layer and the charge transfer between Pb and TaS₂, is absent. Hence, in its current form I do not recommend publication of the manuscript in Nature Communications.

Here are other issues:

1. In Fig. 3c, the authors showed the magnetic-field dependence of the zero-bias peak, which doesn't split up to 8.5 T. It is known that a magnetic field broadens the Kondo peak and eventually splits it at high magnetic field. But there is no clear broadening or suppression of the zero-bias peak in Fig. 3c. It is thus not convincing to specify the zero-bias peak as a Kondo peak.
2. The authors proposed that charge transfer and spatial charge modulation could occur under Pb intercalation in TaS₂. To better support this point, first-principle calculations of the structures of Pb layer under TaS₂ and the resulting charge transfer should be provided.
3. Why does the narrow band of the dark SD cluster look more prominent in the dI/dV than that of the bright SD cluster (Fig. 2d)? What does the dI/dV of the dark region look like for TaS₂ with higher Pb-intercalation concentration (e.g. Figs. 4c and 4d)?
4. Does the dI/dV measured on the Pb layer above the TaS₂ show any signature of a heavy-fermion hybridization gap as in Ref. 20?
5. Typos should be corrected. For example, 'concertation' should be 'concentration'.

Reviewer #3 (Remarks to the Author):

The authors show the emergence of a Kondo peak in a system formed by lead and 1T-TaS₂. The emergence of this Kondo peak stems from the interplay of local moments in 1T-TaS₂, and the metallic states of Pb. The study of Kondo effect in two-dimensional materials is a direction of great interest, and the results of the authors put forward a highly original discovery, establishing the emergence of local magnetic moments in 1T-TaS₂.

The manuscript is very well written, the results are discussed in detail, and they are consistent with well-known results in the field. The authors perform a careful study of the observed phenomenology, using well-established techniques in the field. Their results are presented in great clarity and provide a highly original addition to the field. In particular, I believe that their results are of great interest to the wide community of two-dimensional materials and quantum magnetism.

For the reasons stated above, I strongly recommend their manuscript for publication in Nature Communications.

Point-by-point reply to the reviewers

We thank all the reviewers for their very valuable feedback which has made our paper stronger. We are grateful to the reviewers for pointing out the significance of our results, and we thank Reviewer #3 for his very strong recommendation for publication in *Nature Communications*. We are also grateful for their remarks on where the clarification and improvement are needed. We think we have clarified all the issues raised by the reviewers. All changes are marked in blue in the manuscript.

Best regards,
Shichao Yan on behalf of the authors

Reviewer comments

Reviewer #1 (Remarks to the Author):

In the present manuscript, Shen et al. investigate the lattice Kondo problem by studying the coupling between localized and itinerant electrons in Pb-intercalated correlated insulator 1T-TaS₂. The authors exploit novel Pb intercalation technique to tune the surface layer gradually between insulating and metallic states. In the metallic state, they observe the zero-bias peak (ZBP) in tunneling spectra, which they associate with the Kondo resonance by analyzing its behavior with temperature and magnetic field. The authors use these signatures to prove the local magnetic moment formation on the 1T-TaS₂ surface, and thus support the Mott gap and quantum spin liquid scenarios widely debated recently. These finding also shed light onto the novel many-body regimes associated with the spin degree of freedom. The experimental results presented in the manuscript are convincing, of very high quality and obtained at high technical level. They will certainly be of great interest to the broad community working in the areas of correlated systems and material science.

Response: We thank the reviewer for thoroughly reviewing the manuscript and his very valuable comments which have helped us improve the manuscript significantly. We also thank the reviewer for his positive response to our work.

While I find the results novel and interesting, the manuscript in its present version lacks important pieces of analysis and has somewhat convoluted message. Major revision is necessary to consider it for publication in *Nature Communications*. The detailed comments are provided below.

Major concerns:

1) The manuscript can benefit from a clear picture of the transition from a Mott state to a narrow-band state and Kondo-resonance therein that is suggested to occur in Type-II-terminated surfaces.

1a. Does the Mott state collapse under intercalation? If yes, which process is the driving force – bandwidth, doping, charge transfer etc.?

Response: We thank the reviewer for this important comment. The insulating gap has also been detected in single layer 1T-TaSe₂ or single layer 1T-TaS₂ grown on bilayer graphene (*Nat. Phys.* 17, 1154 (2021) and *Nature* 599, 582 (2021)). When single layer 1T-TaSe₂ (1T-TaS₂) is grown on single layer 1H-TaSe₂ (1H-TaS₂), the Kondo resonance peak appears. This insulating gap to Kondo

resonance transition in single layer 1T-TaSe₂ (1T-TaS₂) has been attributed to be the coupling between the localized spins in the 1T layer and the itinerant electrons in the underneath 1H layer. Motivated by the recent STM results in *Nat. Phys.* 17, 1154 (2021), Chen *et al.* have developed a model of correlated electrons coupled by tunneling to a layer of itinerant metallic electrons (*Phys. Rev. B* 103, 085128 (2021)), which agrees with the STM experiments on single layer 1T-TaSe₂. Chen *et al.* use the below Hamiltonian to describe the system, where “V” is the tunneling amplitude between the correlated electrons and the itinerant electrons.

$$\begin{aligned}
H = & -t_d \sum_{(i,j),\sigma} d_{i,\sigma}^\dagger d_{j,\sigma} + \sum_i n_{d,i} (\epsilon_d^{(0)} - \mu_F) \\
& - t_c \sum_{(i,j),\sigma} c_{i,\sigma}^\dagger c_{j,\sigma} + \sum_i n_{c,i} (\epsilon_c^{(0)} - \mu_F) \\
& + \frac{U}{2} \sum_i (n_{d,i} - 1)^2 - V \sum_{i,\sigma} (c_{i,\sigma}^\dagger d_{i,\sigma} + \text{H.c.})
\end{aligned}$$

We think the situation in our work can also be explained by the model developed by Chen *et al.* Without Pb intercalation, the type-II 1T-TaS₂ layer is on top of the dimerized 1T-TaS₂ which has a ~150 mV insulating gap. In this case, we observe the ~50 mV gap on the type-II 1T-TaS₂ surface. With Pb intercalation, there is charge transfer between the intercalated Pb atoms and the 1T-TaS₂ layers. This creates the electrons that can couple to the localized spins in the type-II 1T-TaS₂ layer and induces the Kondo resonance peak.

To make this point clear, we have added more discussions in the main text. Now it reads as: “Recently, motivated by the Kondo resonance in single layer 1T-TaSe₂ grown on single layer 1H-TaSe₂¹¹, Chen *et al.* have developed a model for the correlated electrons coupled by tunneling to a layer of itinerant metallic electrons to explain the transition to the Kondo resonance²⁸. We think the insulating gap to Kondo resonance transition in Type-II surface can also be explained by the model developed by Chen *et al.*²⁸ The ~50 mV insulating gap in the pristine Type-II surface is previously interpreted as a Mott-insulating gap that is induced by the in-plane Coulomb interaction¹⁹.”

1b. Are the itinerant CDW states affected by intercalation? Here I would like to point the authors’ attention to the paper by Qiao *et al.* *Phys. Rev. X* 7, 041054 (2017) and especially the green curve in Fig. 4e that seems comparable to that seen in dark clusters in the present study. Could the orbital interplay suggested there be responsible for the Mottness collapse in the present case? I would suggest measuring spectroscopic maps at energies corresponding to the expected itinerant band energies to address these questions.

Response: According to our STM data taken on the Pb intercalated Type-II 1T-TaS₂ surface, the intercalated Pb atoms have very little influence on the long-range CDW order. For example, the yellow circles in Fig. 2a indicate the centers of the SD clusters in the Pb intercalated region with dark SD clusters and the Pb-atom-free region enclosed by the black dashed lines. As we can see, the SD clusters in these two regions are well aligned and there is no CDW domain wall between these two regions. This effect can also be seen in Fig. 1g. Although there are randomly distributed dark SD clusters induced by the intercalated Pb atoms, no CDW domain wall can be seen in Fig. 1g.

For the Se substituted 1T-TaS₂ reported by Qiao *et al.* *Phys. Rev. X* 7, 041054 (2017), the

long-range CDW splits into domains as increasing the Se concentration. The difference between the Pb intercalated Type-II surface and the Se substituted 1T-TaS₂ sample is that the intercalated Pb atoms are in the van der Waals gap, which makes their pinning effect to the CDW weaker than that of the substituted Se atoms. Similar effect has also been reported that the long-range CDW order on 1T-TaS₂ is robust when evaporating the potassium atoms on the surface of 1T-TaS₂ (*Phys. Rev. Lett.* 123, 206405 (2019)). We think the origin of the Mottness collapse in the Pb intercalated Type-II surface is likely not the same as that in the Se substituted 1T-TaS₂. We have added this point in the main text and showed more *dI/dV* maps taken on the same region as in Fig. 5a with different bias voltages in Fig. S14.

2) The authors do not provide enough evidence that the dark clusters in Type-II termination are the ones with Pb atom beneath. The arguments provided in the Supplementary Section 3 are logically incomplete. For example, some clusters in Fig. 2 are have spectral intensity somewhere between the bright and dark ones. In my opinion, the smoking-gun evidence would be the tip-induced manipulation of the intercalant atom. Can gentle bias pulsing change the position or appearance of the dark clusters?

Response: In the Supplementary Section 5 and Section 8, we demonstrate that the dark SD clusters are not due to the intrinsic defects in 1T-TaS₂ and are not the Pb adatoms on the surface. Pb atoms also have no chemical reactions with the 1T-TaS₂ surface. Based on these observations, we conclude that the dark SD clusters that can be seen in the STM topography taken with negative bias voltage are the intercalated Pb atoms.

First, as shown in Fig. R1, the similar behavior has also been reported in the Cu intercalated 1T-TiSe₂ material (bulk Cu_xTiSe₂), where the intercalated Cu atoms can be seen as dark spots in STM topography taken with large negative bias voltages (*Phys. Rev. Lett.* 118, 106405 (2017) and *Phys. Rev. Lett.* 118, 017002 (2017)).

Figure R1 STM topographies taken on the same region with different bias voltages on Cu_{0.08}TiSe₂ sample (*Phys. Rev. Lett.* 118, 106405 (2017)). The individual intercalated Cu atoms can be seen as individual dark atomic spots in (b). The number of the dark spots agrees with the nominal doping of the Cu_{0.08}TiSe₂.

Second, intercalation of Pb under graphene has been reported when evaporating Pb onto the graphene/Ir(111) surface (*Nat. Phys.* 11, 43 (2015)), which indicates Pb atoms can be intercalated into the van der Waals gap of two-dimensional materials.

Third, as shown in Fig. R2, it has been reported that the van der Waals gap under the Type-II 1T-TaS₂ surface is slightly larger than that below the Type-I 1T-TaS₂ surface. We think that is the reason that Pb atoms are easier to intercalate into the van der Waals gap below the Type-II 1T-TaS₂ surface.

Figure R2 Step heights for the Type-I and Type-II 1T-TaS₂ layer (*Nat. Commun.* 11, 2477 (2020)).

We completely agree with the reviewer that the smoking-gun evidence would be the manipulation of the intercalated Pb atoms with STM tip. However, manipulating the intercalated atoms below the topmost layer in STM experiments is not an easy task. Although we haven't found a controllable way of manipulating the intercalated atoms, the position of the dark cluster can be changed but rarely during the STM measurements. As shown in the STM topographies in Fig. R3, after performing a few dI/dV measurements in this region, the dark SD cluster marked by the purple circle is changed while the dark SD cluster marked by the green circle remains unchanged. This indicates that the dark SD cluster marked by the purple circle is moved by the STM tip. We have added more discussions about the Pb intercalation and the Fig. R3 in Supplementary Section 4.

Figure R3 STM topographies taken before (a) and after (b) performing the dI/dV measurements in this region (a: $V_s = -200$ mV, $I = 20$ pA; b: $V_s = -200$ mV, $I = 20$ pA). The purple and green circles indicate the positions of two SD clusters.

3) The authors should identify what changes to the long-range charge order are introduced by the Pb intercalation/deposition to put this study into context. Fourier transforms of the topographic images would be very helpful here.

I could not but notice the robustness of the uniform $\sqrt{13} \times \sqrt{13}$ state to the small levels of intercalation, which is seen also in doped and chemically substituted samples. Do the authors see the formation of domain structures or individual domain walls as well (Fig. S3a maybe)? It would

be interesting to match this behavior to other notable cases of the intercalated TMDCs (e.g. 1T-Cu_xTiSe₂) and to simulations of the doped Coulomb lattice gas (cf. Karpov and Brazovskii, *Sci. Rep.* 8, 4043 (2018) and Vodeb et al., *New J. Phys.* 21, 083001 (2019)).

Response: We understand that the reviewer's concern is whether the intercalated Pb atoms create CDW domain walls in 1T-TaS₂. Yes, the CDW domain walls have been reported in the intercalated TMDs materials, such as Cu_xTiSe₂ (*Phys. Rev. Lett.* 118, 106405 (2017)). CDW domain walls can also be created in 1T-TaS₂ by laser pulses or by applying electrical pulses to STM tip. However, the long-range CDW order in 1T-TaS₂ is robust when evaporating individual metal atoms onto the surface of 1T-TaS₂ (*Phys. Rev. Lett.* 123, 206405 (2019)).

According to our STM data taken on the Pb intercalated Type-II 1T-TaS₂ surface, the intercalated Pb atoms also have little influence on the long-range CDW order. For example, the yellow circles in Fig. 2a indicate the centers of the SD clusters in the Pb intercalated region with dark SD clusters and the Pb-atom-free region enclosed by the black dashed lines. As we can see, the SD clusters in these two regions are well aligned and there are no CDW domain walls between them. We think this is due to the weak pinning effect of the intercalated Pb atoms to the CDW order in 1T-TaS₂.

As suggested by the reviewer, we perform Fourier transform (FT) to the STM topographies shown in Figs. 1g and 2a (as shown in the below Fig. R4). The yellow circles in Figs. R4b and R4e indicate the CDW wavevectors. We then perform inverse Fourier transform (IFT) by filtering the CDW wavevectors (Figs. R4c and R4f). As shown in Figs. R4c and R4f, the CDW order in the bare 1T-TaS₂ region is long-range ordered without CDW domain walls.

Figure R4 **a**, STM topography shown in Fig. 1g. **b**, FT of **a**. **c**, The IFT by filtering the CDW wavevectors in the yellow circles in **b**. **d-f**, The same as a-c, but performed for STM topography shown in Fig. 2a. The yellow circles in **b** and **e** indicate the CDW wavevectors. The purple dashed lines in **a**, **c**, **d**, and **e** indicate the single long-range CDW domain region.

However, in the large-area STM topographies taken on both Type-I and Type-II surfaces, we indeed can find a few CDW domain walls (as shown by the yellow dashed lines in the below Figs. R5a and R5b). These CDW domain walls often connect the Pb islands and are likely pinned by these Pb islands. This indicates the Pb islands can induce a few CDW domain walls, but the CDW domain walls are not directly related with the intercalated Pb atoms. On both Type-I and Type-II surfaces with Pb islands, we can find single-CDW-domain regions with the sizes of a few tens of nanometers. We have added the discussions about the influence of the intercalated Pb atoms to the long-range CDW order in the main text and Supplementary Section 1 and Section 7.

Figure R5 a, Constant-current STM topography taken on the Type-I 1T-TaS₂ surface with Pb islands. **b**, Constant-current STM topography taken on the Type-II 1T-TaS₂ surface with Pb islands. The yellow dashed lines indicate the CDW domain walls.

4) In lines 103-104 the authors claim that the 150-mV insulating gap in Type-I termination is understood as the band gap, since Pb intercalation does not affect it. However, no proof of intercalation in Type-I termination is provided, moreover the authors say the surface is indistinguishable from the pristine one. Therefore, the only safe conclusion so far is that intercalation does not occur on the Type-I-terminated surface, rather than the gap is of the band insulator.

Response: We thank the reviewer for this suggestion. Although recently there are several papers that claim the ~150 mV insulating gap is a band insulating gap (*Nat. Commun.* 11, 2477 (2020), *Nat. Commun.* 11, 4215 (2020), *Phys. Rev. Lett.* 126, 256402 (2021), *Phys. Rev. Lett.* 126, 196405 (2021)), the behaviors of this gap related with the correlation effect continue to be reported. In our work, Pb atoms do not intercalate below the Type-I surface and the ~150 mV insulating gap is not affected. We agree with the reviewer that to be more precise, we have deleted the sentence “Thus the system remains in the band insulating state, and the ~150 mV insulating gap is understood as the band gap”.

5) The authors demonstrate that the magnetic field has no effect on the ZBP, consistent with the Kondo resonance picture. The message will be much stronger, if some effect is demonstrated for the narrow band in the dark cluster. Alternatively, an estimate of the expected magnetic field effect should be provided.

Response: We thank the reviewer for raising this important comment. As shown in Fig. 3c, no clear splitting is observed in the Kondo resonance peak with external magnetic field up to 8.5 T. However, for the Kondo resonance peak in the single layer 1T-TaS₂ is split with 10 T magnetic field (Ref. 20). The difference is that the width of Kondo resonance peak in Fig. 3c is about four times the width of the Kondo resonance peak in the single layer 1T-TaS₂ (Fig. R6). This makes the Kondo resonance peak in Fig. 3c less sensitive to the splitting induced by the magnetic field, and larger magnetic field is needed to induce the splitting of the Kondo resonance in the Pb intercalated Type-II 1T-TaS₂ surface. We have added the discussion in the main text and Fig. R6 in Supplementary Section 10.

Figure R6 The dI/dV spectra taken with 0 T (blue) and 8.5 T (red) magnetic fields shown in Fig. 3c. The dI/dV spectra taken on the 1T-TaS₂ layer in the 1T/1H heterostructure with 0 T (purple) and 10 T (green) magnetic fields (from Ref. 20).

Motivated by the reviewer's comment, we also find that similar effect has also been observed in the heavy fermion system. As shown in Fig. R7 from Ref. 27, the Kondo lattice peak in the heavy-fermion metal YRh₂Si₂ has no clear splitting even with magnetic field up to 11 T.

Figure R7 Magnetic field dependent dI/dV spectra taken on the heavy-fermion metal YRh₂Si₂ (from Ref. 27).

6) Could the authors comment on the spatial variation of the ZBP width (Fig. 2)? Furthermore, is the temperature dependence shown in Fig. 3a measured at the same spot?

Response: We showed the ZBP in Figs. 2d and 2e. In Fig. 2d, the purple spectrum taken on the bright SD cluster shows strong ZBP and the it is strongly suppressed in the green dI/dV spectrum

taken on the dark SD cluster. At the same time, the narrow electronic band located slightly above the Fermi level appears in the green dI/dV spectrum in Fig. 2a. This indicates that when the narrow electronic band is located slightly above the Fermi level, the ZBP is strongly reduced.

Fig. 2e shows the dI/dV spectra taken along the red arrow shown in Fig. 2c. In order to clearly show the spatial variation of the ZBP shown in Fig. 2c, we have fitted and plotted the width of the ZBP in each dI/dV spectrum shown in Fig. 2c. As we can see from Fig. R8, the width of the ZBPs in the bright SD clusters has very little spatial variation. We have added Fig. R8 in the Supplementary Section 9.

Figure R8 a, dI/dV spectra shown in Fig. 2e. The spectra are vertically offset for clarity. The red lines are the thermal convolved Fano fits to the ZBP. **b**, The half width (Γ) at half maximum of the dI/dV spectra shown in (a). The red dashed line is the averaged Γ .

Yes, the temperature dependent dI/dV spectra shown in Fig. 3a are measured at the same spot. At each temperature, we find the same area and the same spot to perform the dI/dV measurements. To make this point clear, we have added the STM topography containing the spot for the temperature dependent dI/dV measurements in the inset of the new version of Fig. 3a. In order to be consistent with Kondo fitting used in Ref. 23, we use the thermally convolved Fano fits to fit the Kondo resonance peak in the new Fig. 3.

Minor concerns:

1) The authors find Type-I and Type-II terminations in a room-temperature cleaved crystal, in contrast to the recipe and reasoning provided in Ref. 13. I believe this observation deserves a comment, at least in the supplement.

Response: Yes, we cleave the 1T-TaS₂ sample and evaporate Pb atoms on the 1T-TaS₂ surface at room temperature. We find the Type-I and Type-II surface terminations with Pb islands. As reported in Ref. 13, the Type-I surface is the most common cleaved surface, and they suggest that “the maintenance of the temperature far below ~180 K is important for achieving cleaved surfaces that yield information about the pre-formed bulk stacking structure”. We think the reason that we can observe the two surface terminations in the room-temperature cleaved

sample is due to the Pb atoms that could disturb the stacking between the topmost 1T-TaS₂ and the underneath 1T-TaS₂. This could be helpful for the formation of the Type-II surface. We have added this discussion in Supplementary Section 1.

2) The authors nicely demonstrate the feasibility of the intercalation as the tuning knob in 1T-TaS₂. I believe a more detailed recipe should be provided such that others could exploit this technique as well.

Response: As suggested by the reviewer, we have added more detailed information about evaporating Pb atoms in the method part.

3) Is there a way to quantify the local or mean intercalant concentration? Could the authors match the annealing time to intercalant concentration?

Response: Yes, since the dark SD clusters are induced by the individual intercalated Pb atoms, we think the simple method of quantifying the intercalant concentration is counting the density of dark SD clusters in the STM topography.

The intercalant concentration is indeed related with the annealing time. As shown in the below STM topographies, we can see the individual randomly distributed dark SD clusters with 5 minutes room-temperature annealing. When we perform room-temperature annealing for 10 minutes, the density of the dark SD clusters becomes higher. However, after 10 minutes annealing, the spatial distribution of the dark SD clusters is not very uniform. As shown in Fig. R9, we can find different intercalant concentration of Pb in different areas. We have added the STM topographies in Supplementary Section 12.

Figure R9 STM topographies taken with 5 min (a) and 10 min (b) room-temperature annealing.

4) Intercalation conditions, at which the images shown in panels a-d in Figure 4, should be provided.

Response: As we mentioned above, after 10 minutes room-temperature annealing, the spatial distribution of the intercalated Pb atoms is not spatially uniform. We can find different intercalant concentration. We have added this in the Supplementary Section 12.

5) Could the authors comment on the very different appearance of the Pb islands on Type-I and Type-II terminations? Are there domain walls on Type-I termination? Could the periodic height modulation on Type-II termination be related to a strain pattern or reconstruction?

Response: This is indeed a very interesting question. Yes, as shown in the Fig. S2, the Pb islands appear different on Type-I surface and Pb intercalated Type-II surface. We think that is because of the intercalated Pb atoms that make the Type-II surface become metallic. As we can also see from the below Fig. R10 and Fig. 2a in the main text, for the Type-II surface, when the Pb islands are grown on the small region where there are no dark SD clusters and with ~ 50 mV insulating gap, the Pb islands appear similar as that on the Type-I surface. It is very likely that the Pb islands have different structures when they are grown on the metallic and the insulating 1T-TaS₂ surfaces. It is possible that the periodic height modulation in the Pb islands is due to the strain pattern. We have added the discussions in Supplementary Section 2.

Yes, after forming the Pb islands, a few CDW domain walls can be found on both the Type-I and Type-II surfaces, which can be seen in Fig. S1 in the Supplementary Section 1. The CDW domain walls are likely pinned by the Pb islands, and they are not directly related with the intercalated Pb atoms. We have added this comment in the Supplementary Section 2.

Figure R10 Constant-current STM topography taken on the Type-II 1T-TaS₂ surface with Pb islands.

Technical comments:

a. Line 45: “...undergoes a commensurate CDW transition around 180K...” – I suggest using 180-190K, as this would be more consistent with the cited sources and literature data.

Response: As suggested by the reviewer, we have changed the temperature as “180-190 K” to be consistent with the literature data.

b. Line 53: “...emergent triangular lattice...” – what property is emergent here?

Response: In the previous version, we use “emergent triangular lattice” to emphasize the SD cluster lattice has different periodicity with the Ta atoms. We think we have already described the periodicity of the SD cluster as “*In this commensurate CDW state of 1T-TaS₂, the in-plane lattice distortion leads to the formation of Star-of-David (SD) cluster which consists of 13 Ta atoms*”. We have deleted the sentence with “emergent triangular lattice” in the main text.

c. Line 58: reference 11 describes an epitaxially-grown monolayer of 1T-TaSe₂ with the physics quite different from the bulk 1T-TaSe₂.

Response: As mentioned by the reviewer, to be more accurate, we have changed the sentence to as “... the scanning tunneling microscopy (STM) measurement in the epitaxially-grown isostructural material (single layer 1T-TaSe₂)¹¹.”

d. Line 138: “...the dark SD cluster in Fig. 2c” – please, add ‘green arrow’ to enhance the readability of the paper.

Response: Thank you for this suggestion. We have added “marked by the green arrow in Fig. 2c” in the main text.

Reviewer #2 (Remarks to the Author):

Shiwei et. al. studied the effect of Pb intercalation on bulk 1T-TaS₂ using STM and STS. They showed that for certain amount of Pb intercalation, the insulating gap in the pristine Type-II 1T-TaS₂ surface becomes a zero-bias peak in the dI/dV spectra, which is claimed to be a Kondo resonance peak.

My major concern is about the novelty of the work for the Nature Communications standards. Kondo resonance in Ta-based dichalcogenide such as 1T-TaSe₂ and 1T-TaS₂ have been reported recently as in *Nat. Phys.* 17, 1154 (2021) and arXiv:2103.11989 (2021). In addition, theoretical calculations to justify the main claim of the paper, such as the modeling of structural properties Pb intercalation layer and the charge transfer between Pb and TaS₂, is absent. Hence, in its current form I do not recommend publication of the manuscript in Nature Communications.

Response: We thank the reviewer for his valuable comments and the suggestion for adding the theoretical calculations. About the novelty of our work, although the Kondo effect has been very recently reported in epitaxially-grown monolayer 1T-TaSe₂ (*Nat. Phys.* 17, 1154 (2021)) and monolayer 1T-TaS₂ (*Nature* 599, 582 (2021)), this effect has not been realized in the bulk 1T-TaS₂ (or bulk 1T-TaSe₂) sample, and the relationship between the Kondo resonance and the narrow electronic band formed by the unpaired electrons in the SD clusters has not been studied at all. In our work, we not only demonstrate that Pb intercalation can induce the insulating gap to Kondo resonance transition in the Type-II 1T-TaS₂ surface, but also clearly show that Pb intercalation can be used as a knob to tune the Kondo resonance by shifting the position of the narrow electronic band. As commented by the reviewer #3, our results should be of great interest to the wide community of two-dimensional materials and quantum magnetism.

We also note that when we submitted our manuscript to *Nature Communications*, the Kondo resonance paper about single layer 1T-TaS₂ (arXiv:2103.11989 (2021)) has not been formally published yet. It is published in *Nature* on November 24th, 2021 (*Nature* 599, 582 (2021)) which is after we received the review reports from *Nature Communications*.

Here are other issues:

1. In Fig. 3c, the authors showed the magnetic-field dependence of the zero-bias peak, which doesn't split up to 8.5 T. It is known that a magnetic field broadens the Kondo peak and eventually splits it at high magnetic field. But there is no clear broadening or suppression of the zero-bias peak in Fig. 3c. It is thus not convincing to specify the zero-bias peak as a Kondo peak.

Response: We thank the reviewer for this important comment about the magnetic field

dependence of the zero-bias peak. The reviewer is correct that when the magnetic field is strong enough, it will broaden and eventually split the Kondo peak. For example, with 10 T magnetic field, the splitting is observed for the Kondo resonance peak in the single layer 1T-TaS₂ grown on 1H-TaS₂ layer (Ref. 20). However, as shown in the below Fig. R11, the width of Kondo resonance peak in Fig. 3c is about four times the width of the Kondo resonance peak in the single layer 1T-TaS₂. This makes the Kondo resonance peak in Fig. 3c less sensitive to the splitting induced by the magnetic field, and larger magnetic field is needed to induce the splitting of the Kondo resonance peak in the Pb intercalated Type-II 1T-TaS₂ surface. We have added the discussion in the main text and Supplementary Section 10.

Figure R11 The dI/dV spectra taken with 0 T (blue) and 8.5 T (red) magnetic fields shown in Fig. 3c. The dI/dV spectra taken on the 1T-TaS₂ layer in the 1T/1H heterostructure with 0 T (purple) and 10 T (green) magnetic fields (from Ref. 20).

Motivated by the reviewer's comment, we also find that similar effect has also been observed in the heavy fermion system. As shown in the below Fig. R12 from Ref.27, the Kondo lattice peak in the heavy-fermion metal YRh₂Si₂ has no clear splitting with magnetic field up to 11 T.

Figure R12 Magnetic field dependent dI/dV spectra taken on the heavy-fermion metal YRh₂Si₂ (from Ref. 27).

2. The authors proposed that charge transfer and spatial charge modulation could occur under Pb intercalation in TaS₂. To better support this point, first-principle calculations of the structures of Pb layer under TaS₂ and the resulting charge transfer should be provided.

Response: We agree with the reviewer that it would be more convincing to add the

first-principles calculations. We have performed the first-principles calculations which show that there is indeed charge transfer between the intercalated Pb atoms and the adjacent 1T-TaS₂ layers.

As shown in the below Fig. R13, the charge analysis shows that ~ 1.1 electrons transfer from the Pb atom to the adjacent 1T-TaS₂ layers. For the intercalated Pb atom below the Type-II surface, the transferred electrons are symmetrically distributed in two adjacent layers. The transferred electrons from the intercalated Pb atom are mainly redistributed at the neighboring Ta and S atoms (Fig. R13 (c)). We have added the first-principles calculations in the Supplementary Section 11.

Figure R13 a and b, Side view of charge densities for 1T-TaS₂ (a) and Pb intercalated below the Type-II 1T-TaS₂ surface (b). Isosurface value is 0.1 e/bohr³. **c**, Side view of differential charge densities for the intercalated Pb atom below the Type-II 1T-TaS₂ surface. Isosurface value is 0.002 e/bohr³. Green and purple contours indicate charge accumulation and reduction, respectively. **d**, Schematic diagram of the charge transfer of Pb atom to the adjacent 1T-TaS₂ layers. Here for convenient view only Ta atoms are shown in the corresponding 1T-TaS₂ layers.

3. Why does the narrow band of the dark SD cluster look more prominent in the dI/dV than that of the bright SD cluster (Fig. 2d)? What does the dI/dV of the dark region look like for TaS₂ with higher Pb-intercalation concentration (e.g. Figs. 4c and 4d)?

Response: In the bright SD cluster, the Kondo resonance peak is very strong which makes the narrow electronic band near the Fermi level less prominent. However, in the dark SD cluster, the narrow band is locally shifted to be slightly above the Fermi level, which decreases the Kondo coupling between the localized electrons in the narrow band and the itinerant electrons. In this case, the Kondo resonance peak is suppressed, and the narrow band appears more prominent in the dark SD cluster. We have added this point in the main text.

We agree with the reviewer that we should show the evolution of the dI/dV spectra in the dark SD cluster region as increasing the Pb intercalation concentration. We have plotted the evolution of the dI/dV spectra in the dark spot in Fig. R14 and added it in Supplementary Section 13.

Figure R14 a-d, Constant-current STM topographies taken on the Type-II 1T-TaS₂ surface as increasing the intercalant concentration of Pb. **e**, dI/dV spectra taken on the colored arrows shown in (a-d).

4. Does the dI/dV measured on the Pb layer above the TaS₂ show any signature of a heavy-fermion hybridization gap as in Ref. 20?

Response: We thank the reviewer for this comment. As shown in the below Fig. R15, we observe the dip-like features near the Fermi level in the Pb islands grown on both Type-I and Type-II 1T-TaS₂ surfaces. These dip-like features can also be seen in Figs. 1f and 1i. The Pb islands grown on the Type-II surface have smaller gap than the Pb islands grown on the Type-I surface. However, we cannot confidently identify the gap in the Pb islands on the Type-II surface as the heavy-fermion hybridization gap.

Figure R15 The typical dI/dV spectra taken on the Pb islands grown on the Type-I (red) and Type-II (blue) surfaces.

5. Typos should be corrected. For example, ‘concertation’ should be ‘concentration’.

Response: Thank you and we are sorry for the typos. We have changed the “concertation” to be “concentration” and “inserts” to be “insets”.

Reviewer #3 (Remarks to the Author):

The authors show the emergence of a Kondo peak in a system formed by lead and 1T-TaS₂. The

emergence of this Kondo peak stems from the interplay of local moments in 1T-TaS₂, and the metallic states of Pb. The study of Kondo effect in two-dimensional materials is a direction of great interest, and the results of the authors put forward a highly original discovery, establishing the emergence of local magnetic moments in 1T-TaS₂.

The manuscript is very well written, the results are discussed in detail, and they are consistent with well-known results in the field. The authors perform a careful study of the observed phenomenology, using well-established techniques in the field. Their results are presented in great clarity and provide a highly original addition to the field. In particular, I believe that their results are of great interest to the wide community of two-dimensional materials and quantum magnetism.

For the reasons stated above, I strongly recommend their manuscript for publication in Nature Communications.

Response: We thank the reviewer for his very positive response to our work and the strong recommendation for the publication.

REVIEWER COMMENTS

Reviewer #1 (Remarks to the Author):

The authors have revised the manuscript addressed the raised criticism carefully and in details. The manuscript has substantially improved. I am satisfied with the response, and I would be happy to recommend the manuscript after the following point is resolved:

In response to the question 1a of the original review on the origin of the Mottness collapse, the authors suggested the mechanism described by Chen et al., (PRB 103, 085128 (2021)) is responsible. I believe such claims require sanity check, in particular, what kind of trajectory on the theoretical phase diagram does the system experience upon Pb intercalation? Since Chen et al. provide some recipes to estimate the important parameters from STS measurements, I suggest the authors extract those from their data and see how realistic the scenario is then. In addition, Chen et al. predict certain temperature dependence of the Kondo peak width in different regimes. Can the authors compare their data with the theory?

Reviewer #2 (Remarks to the Author):

The authors have addressed my concerns in the revised manuscript and I recommend publication.

Point-by-point reply to the reviewers

REVIEWER COMMENTS

Reviewer #1 (Remarks to the Author):

The authors have revised the manuscript addressed the raised criticism carefully and in details. The manuscript has substantially improved. I am satisfied with the response, and I would be happy to recommend the manuscript after the following point is resolved:

Response: We thank the reviewer for his positive response to our revised manuscript. We are also pleased to read that the reviewer thinks our manuscript has been substantially improved. We thank the reviewer again for his very valuable comments.

In response to the question 1a of the original review on the origin of the Mottness collapse, the authors suggested the mechanism described by Chen *et al.*, (PRB 103, 085128 (2021)) is responsible. I believe such claims require sanity check, in particular, what kind of trajectory on the theoretical phase diagram does the system experience upon Pb intercalation? Since Chen *et al.* provide some recipes to estimate the important parameters from STS measurements, I suggest the authors extract those from their data and see how realistic the scenario is then. In addition, Chen *et al.* predict certain temperature dependence of the Kondo peak width in different regimes. Can the authors compare their data with the theory?

Response: We thank the reviewer this comment. In Ref. 28 (*Phys. Rev. B* 103, 085128 (2021)), Chen *et al.* study a model of correlated electrons coupled by tunneling to a layer of itinerant metallic electrons. They also provide the below phase diagram (Fig. R1) for their model system. By tuning the parameters, Chen *et al.* compare and fit the previously reported temperature-dependent Kondo resonance width in the 1T/1H 1T-TaSe₂ heterostructure with their theoretical results (Ref. 11). They conclude that the fit deteriorates as t_d/U increases, especially when it is away from the orange dashed line shown in Fig. R1.

Fig. R1 Phase diagram taken from Fig. 5 in Ref. 28.

Chen *et al.* also find that the theoretical and experimental results match with each other in the Anderson limit ($t_d/U = 0$) or in the intermediate t_d/U (Figs. 11 and 12 in Ref. 28). In these cases, the theoretical results can be well fitted by $\Gamma = \sqrt{a\pi^2(k_B T)^2 + (\Gamma_0)^2}$ with a close to 1 (Eq. 32 in Ref. 28). As shown by the red solid line in Fig. 3b in our manuscript, the temperature-dependent width of the Kondo peak can also be well-fitted with the equation $\sqrt{(\pi k_B T)^2 + 2(k_B T_K)^2}$ which is equivalent to the Eq. 32 in Ref. 28 with $a = 1$. This indicates by adjusting the parameters, our experimental results in Fig. 3b should also be able to be fitted with the model in Ref. 28 in the Anderson limit ($t_d/U = 0$) or in the intermediate t_d/U .

As shown in Fig. R1, without the coupling to the itinerant metallic electrons, the Type-II 1T-TaS₂ layer should be in the spin liquid phase. In comparison with the 1T/1H 1T-TaSe₂ heterostructure in Ref. 28, the situation for the Type-II 1T-TaS₂ after Pb intercalation is also close to the localized Kondo limit or in an intermediate coupling regime in which the Heisenberg exchange interaction and the Kondo coupling are comparable.

We have added this point in the manuscript. It reads as: “According to the temperature-dependent Kondo resonance width shown in Fig. 3b and the model developed by Chen *et al.* in Ref. 28, the situation after Pb intercalation should be either near the localized Kondo limit or in the regime where the Kondo exchange coupling is comparable with the Heisenberg exchange coupling between the local moments in the 1T-TaS₂ layer²⁸.”

Fig. R2 Calculated width of the Kondo peak taken from Fig. 12c in Ref. 28 (black circles) and the width of the Kondo peak taken from Fig. 3b (blue squares).

As suggested by the reviewer, in Fig. R2, we plot the calculated temperature dependence of the Kondo peak (Fig. 12c in Ref. 28) together with our experimental results in Fig. 3b. As shown in Fig. R2, the experimental and theoretical results match reasonably well at high temperature, but they do not match very well at very low temperature. This indicates based on the parameters used in Fig. 12c of Ref. 28, the parameters need to be further optimized to completely fit our experimental results.

Reviewer #2 (Remarks to the Author):

The authors have addressed my concerns in the revised manuscript and I recommend publication.

Response: We thank the reviewer for his recommendation for publication.

List of changes:

1. To make the abstract to be within 150 words, we deleted the sentence “*and thus confirms the cluster Mott localization of the unpaired electrons and local moment formation in the 1T-TaS₂ layer*”. Now there are ~148 words in the abstract.
2. On page 5, we added the description about the situation after Pb interaction.
3. On page 6, to be more precise, we changed the “*new electronic states*” to be “*many-body electronic states*”.
4. To be clear, we rewrote a few words in the Methods part.

REVIEWERS' COMMENTS

Reviewer #1 (Remarks to the Author):

The authors have addressed my questions and further revised the manuscript. I am happy to recommend this excellent manuscript for publication in Nature Communications.

As a side note to the analysis presented in the authors' response, I would be surprised to find 1T-TaS₂/Pb in the Anderson limit. Nevertheless, I appreciate that the authors discuss both scenarios and leave it for the reader to decide.

Point-by-point reply to the reviewers

REVIEWER COMMENTS

Reviewer #1 (Remarks to the Author):

The authors have addressed my questions and further revised the manuscript. I am happy to recommend this excellent manuscript for publication in Nature Communications.

As a side note to the analysis presented in the authors' response, I would be surprised to find 1T-TaS₂/Pb in the Anderson limit. Nevertheless, I appreciate that the authors discuss both scenarios and leave it for the reader to decide.

Response: We thank the reviewer for the recommendation and all his valuable comments.